# Decentralized Manufacturing Management Based on Federated Learning with Stacking Ensemble

## Abstract

We propose a new intelligent management system to overcome the limitations of privacy, security, communication efficiency, and real-time analysis of data generated in smart manufacturing environments. As the digital transformation of the manufacturing industry accelerates, the importance of data utilization has grown, but the existing centralized approach involves data leakage risk and network load issues. To overcome these limitations, we propose a three-layer federated learning architecture consisting of cloud–anchor–edge. In particular, the anchor layer applies a stacking ensemble technique that combines predictions from multiple models to accurately identify complex anomaly patterns that are difficult to detect with a single model and maximize the robustness of model predictions. Compared to the accuracy of 0.5585 achieved by a single 1D-CNN model, the model applying stacking to federated learning significantly improved performance to an accuracy of 0.7438. Furthermore, to address the continuously changing data distributions in manufacturing environments, we propose a data distribution change detection and edge reallocation mechanism to enhance system flexibility and adaptability. The proposed system demonstrates significantly faster inference times than centralized learning models, presenting it a a powerful alternative that ensures data privacy.

## 1 Introduction

Recently, the global manufacturing industry has accelerated its transition to smart factories, making the efficient collection, analysis, and utilization of real-time data a key factor in corporate competition. This data, which goes beyond simple production records, forms the foundation for innovations like quality control, predictive maintenance, and process optimization. For example, real-time analysis of equipment sensor signals can enable early detection of anomalies, while data-driven redesign of production flows can maximize efficiency and reduce defect rates (Abdullahi et al., 2024). However, this valuable data is also highly sensitive, containing proprietary information such as unique manufacturing processes, core technologies, and supply chain patterns (Lee et al., 2024a). This requires robust privacy and security measures, especially with the rise of international data protection regulations like GDPR and the increasing threat of industrial espionage (Campanile et al., 2021).

Traditional centralized data processing methods, while simple in management and analysis, face significant structural limitations (Oh et al., 2024). They create a single point of failure where the concentration of all data makes them highly vulnerable to security breaches (Menegatti et al., 2023). Furthermore, the transmission of large volumes of data from numerous edge devices to a central server increases network load and reduces real-time performance, which is a major drawback for time-sensitive manufacturing operations (Ahn et al., 2023; Li & et al., 2022). These limitations indicate a critical need for a new data processing paradigm that can safely and efficiently utilize distributed data resources without exposing sensitive information (Daza et al., 2023). The manufacturing sector has seen increased interest in technologies like federated learning, distributed artificial intelligence, and privacy-preserving learning to address these issues.

**C1. How can we ensure data privacy and real-time performance while managing large-scale, distributed manufacturing data?** Traditional centralized methods are not equipped to handle the balance between data utility and security. Transmitting sensitive raw data to a central server creates

significant privacy risks and incurs high communication costs, which limits scalability and real-time responsiveness.

**C2. How can a model adapt to continuously changing data distributions and complex anomaly patterns in a dynamic manufacturing environment?** Manufacturing data is highly dynamic, and factors like equipment maintenance can cause a sharp decline in model performance over time. Moreover, the coexistence of diverse defect types and rare anomalies makes it challenging for a single global model to achieve high detection accuracy.

**Solution to C1. A Hierarchical Federated Learning Architecture.** We propose a novel Decentralized Manufacturing Management System (DMMS) built on a three-layer FL architecture consisting of cloud, anchor, and edge layers. This structure allows the edge layer to perform local model training on-site without exposing sensitive raw data, thereby ensuring data privacy. The anchor layer then efficiently aggregates model weights from multiple edges, improving communication efficiency and enabling flexible scalability.

**Solution to C2. Ensemble Learning and Adaptive Mechanisms.** To address complex anomaly patterns, our anchor layer applies a stacking ensemble technique that integrates predictions from multiple models. This approach significantly enhances defect detection accuracy and generalization performance. Furthermore, we introduce a lightweight adaptation mechanism that detects changes in edge data distribution using the Wasserstein distance and reallocates the edge to the most suitable anchor. This mechanism minimizes performance degradation and ensures long-term stability.

In summary, this study proposes an Decentralized Manufacturing Management System (DMMS) that addresses the limitations of traditional federated learning (FL) in dynamic manufacturing environments. The system uses a novel three-layer FL architecture consisting of cloud, anchor, and edge layers. The edge layer ensures data privacy by training models locally on sensitive data, while the anchor layer efficiently aggregates these models and adapts to changes. A key innovation is the stacking ensemble technique, which improves anomaly detection accuracy by combining predictions from multiple anchor models, making it possible to identify complex and rare defects. Additionally, the system features an adaptation mechanism that detects data distribution shifts and reallocates edge devices to optimal anchors, ensuring long-term model stability and performance. The proposed IMMS simultaneously achieves three key objectives: data privacy, high-precision anomaly detection, and adaptability, providing a robust solution for a wide range of manufacturing sites.

The main contributions of our work are summarized as follows:

- We demonstrate that applying a stacking ensemble to an FL framework significantly improves anomaly detection accuracy, achieving 0.7438, a 33% improvement over the standard 1D-CNN FL method (0.5585).
- Our proposed system shows a substantial reduction in inference time, with the stacking model being approximately 6.3 times faster than a centralized random forest model (2.3882 ms vs. 15.1691 ms), highlighting its suitability for real-time applications.
- We propose a hierarchical FL structure that ensures data privacy and an adaptive mechanism for continuous, stable operation in dynamic environments. To enhance adaptability, we propose a mechanism that uses the Wasserstein distance to detect changes in data distribution and reallocates edges to the optimal anchor using SYN, SYN-ACK, and ACK protocols.

## 2 RELATED WORK

The retrieval and discovery of related work were facilitated using the Gemini large language model to search for academic papers and relevant literature based on specific keywords.

### 2.1 MANUFACTURING MANAGEMENT SYSTEM

A manufacturing management system (MMS) is an information system that manages core tasks across all aspects of manufacturing, including production planning, materials and inventory management, quality control, and equipment maintenance (Dey Sarkar et al., 2024). An MMS collects and analyzes various data generated on the production floor in real time, optimizing production

process flows, reducing unnecessary waste, and improving quality to enhance a company's competitiveness and profitability (Qi & Tao, 2019). Such systems support the effective resolution of various challenges faced by manufacturers, including production schedule automation, work instruction standardization, inventory level optimization, equipment utilization rate maximization, and rapid response to quality issues (Kong et al., 2022). The introduction of MMS increases visibility and transparency in manufacturing sites, promotes standardization and automation of business processes, and leads to various results such as company-wide cost reduction, quality innovation, and improved delivery compliance rates (Mahfoud et al., 2024). In particular, in smart manufacturing environments, various tasks such as production planning and control, quality innovation, energy efficiency, and resource optimization are required simultaneously, necessitating the adoption of integrated management systems based on digital twins, artificial intelligence, IoT, and simulation (Zhu et al., 2024). These changes are expected to strengthen the competitiveness of the manufacturing industry and achieve sustainable growth (Quy et al., 2022).

### 2.2 FEDERATED LEARNING

FL is a distributed machine learning paradigm in which multiple institutions or nodes collaborate to train artificial intelligence models while ensuring data privacy and security (Lee et al., 2024b; Quan & et al., 2025). Each participant trains the model locally without transmitting the original data externally and then shares only the parameters or updates with the central server to improve the overall model performance (Wu et al., 2024). Such FL is categorized into horizontal federated learning (HFL) and vertical federated learning (VFL) based on the data partitioning method (Zheng et al., 2025). The HFL method is applied when multiple institutions share the same feature space but possess different samples (Sah et al., 2025). This approach can ensure data privacy in large-scale distributed environments while improving the generalization performance of the model (Guan et al., 2024). On the other hand, VFL is used when multiple institutions hold data on the same sample but with different characteristics (Leng et al., 2025). In VFL, the involved institutions combine their different characteristics to jointly train the model, while the original data are not shared externally. In this process, the roles are divided between the institution that holds the label information and the institution that provides additional characteristic information. Each institution trains a partial model and exchanges only the intermediate results to complete the model.

### 2.3 STACKING ENSEMBLE METHOD

Stacking is a two-stage learning structure in which multiple base learners independently learn and predict the same input data, and their prediction results are combined to create new data features to be used by a meta-learner to perform the final prediction. The key point here is that the meta-learners treat the predictions of the base learners as a single "input dataset" and comprehensively learn from it to derive the optimal prediction results (Wolpert, 1992; Breiman, 1996). The base learners use different algorithms or model structures to capture various patterns and features of the data (Mienye & Sun, 2022). In other words, they do not simply sum the predictions of the base learners but determine which base learner is more reliable for a specific input or which combination produces the optimal result. This enables stacking to effectively model complex nonlinear dependencies and high-dimensional interactions (Büyükçakir et al., 2018; Mienye & Sun, 2022). Among model ensembles, stacking differs from traditional bagging and boosting in that it refers to a "learned combination." Bagging creates multiple basic learning models in parallel and averages them, while boosting is a sequential learning structure that compensates for the errors of the previous model. In contrast, stacking re-trains the predictions obtained from multiple models using a separate learning model to discover the optimal combination method.

## 3 METHODOLOGY

In the manufacturing industry, the ability to quickly analyze and utilize the vast amounts of data generated by sensors and IoT devices in real time has emerged as a key competitive advantage. These data are essential for smart manufacturing innovations such as quality control, predictive maintenance, and process optimization. At the same time, such information is a valuable asset and can be sensitive for companies, resulting in need for data privacy and security requirements. In this environment, traditional centralized data processing methods have exposed several limitations, in-

cluding risks of data privacy breaches, significant network traffic burdens, and a lack of flexibility in adapting to the diverse and ever-changing manufacturing environment. Therefore, we propose a hierarchical manufacturing management system based on FL utilizing a three-tiered structure of cloud–anchor–edge to safely protect on-site data while efficiently leveraging distributed data resources for learning.

## 3.1 OVERALL ARCHITECTURE

Fig. 1 shows the overall architecture of the proposed DMMS. This system adopts a hierarchical structure based on FL and applies the stacking ensemble technique to enable efficient utilization of the vast amount of data generated in manufacturing sites while ensuring privacy. Each layer is categorized into cloud, anchor, or edge.

- Edge: The edge layer is located at the far end of the system and is the location of data collection directly from sensors and equipment on the manufacturing floor. In accordance with FL principles, the edge layer updates models using local data and evaluates their performance independently. It also detects changes in data distribution and notifies the anchor for transfer to the appropriate location. The edge transmits only the model weights to the upper anchor layer, ensuring strict data privacy.

- Anchor: The anchor layer acts as an intermediate hub that aggregates model weights transmitted from multiple edge devices and performs additional learning. Each anchor updates its own model by integrating parameters collected from multiple edges and communicates with other anchors as needed to continuously improve model performance. This layer is flexibly responds to system expansion, such as client relocation or addition of new anchors, in accordance with changes in data distribution. Finally, the stacking ensemble technique, which uses the prediction results of multiple anchor models as input, performs final anomaly detection and classification to maximize prediction accuracy.

- Cloud: The cloud layer acts as an auxiliary management server for the entire system, focusing on coordinating the flow of information between anchors and edges and monitoring the overall status of the system rather than direct model training. The cloud minimizes centralized control and maximizes the efficiency and scalability of distributed learning by granting autonomy to the anchor and edge layers.

The greatest strength of this hierarchical structure lies in data privacy preservation. Raw data are contained at each edge, and only model parameters are shared with the anchor, strengthening the security of sensitive manufacturing data and facilitating regulatory compliance. Through intermediate aggregation and specialized learning via the anchor layer, various abnormal states can be classified more accurately, and each anchor, specialized by specific defect types, can effectively identify subtle pattern differences that are difficult to capture with a single model. The system also offers excellent scalability and adaptability, allowing easy expansion by adding anchors or reconfiguring existing models when new defect types emerge. Additionally, the hierarchical structure enables efficient communication patterns through the anchor without direct communication between the edge and the cloud, significantly improving overall communication efficiency.

## 3.2 MODEL TRAINING WITH STACKING ENSEMBLE

Algorithm 1 represents the model training and classification process. First, the algorithm repeatedly performs the FL process to integrate the knowledge of distributed edge devices. In each training round, all edges obtain the current model weights from the anchor. The edge trains this model using its local dataset and calculates the updated local weights. In this process, the raw data does not leave the edge, and only the updated weights are uploaded to the upper anchor layer. Subsequently, each anchor aggregates the weight updates received from all edges and updates the anchor model weights using FL algorithms such as FedAvg and Scaffold, reflecting the contributions of all edges to improve the model generalization performance. Once FL is complete over a total of T rounds, a stacking ensemble is constructed using the trained anchor models. This step involves combining the prediction results of the models to perform the final anomaly classification. First, the prediction probabilities of all anchor models are calculated for each sample in the validation dataset, and these probabilities are collected to generate meta features for use as input for meta models such as

XGBoost and are trained. Finally, the trained meta model performs the final anomaly classification prediction based on the meta features of the test dataset. By combining the predictions of multiple anchor models using the stacking ensemble technique, it is possible to accurately identify even complex patterns that are difficult to detect with a single model, maximizing the overall prediction accuracy of the system.

In addition, the system uses a 1D-CNN model for learning, as most sensor data generated in manufacturing sites are in time-series form. This model is optimized to process such time-series data and was selected as the local edge model for the proposed system. 1D-CNN automatically learns important features of time-series data using filters, enabling it to identify recurring patterns or anomalies in the temporal data flow without requiring manual feature design. The 1D-CNN model has a relatively simple structure, resulting in lower computational requirements that are advantageous for efficiently training models on edge devices with low power and limited computing resources. Sensor data from manufacturing sites contain ample noise. The convolution operation of 1D-CNN summarizes the local features of the entire data set and is less affected by noise, allowing clear detection of anomalies.

---

**Algorithm 1** Training and Classification with Stacking

---

1: **Initialization**
2: Initialize $M_{anchors}$ anchor models $\{W_j\}_{j=1}^{M_{anchors}}$.

3: **Federated Learning**
4: **for** $t = 1$ to $T$ rounds **do**
5:     Each client $i = 1 \ldots N_{clients}$ downloads the anchor model $W_j^{(t-1)}$.
6:     Each client trains a local model on its local dataset $C_i$:
7:         $w_{i,j}^{(t)} \leftarrow \text{LocalTrain}(W_j^{(t-1)}, C_i)$
8:     Each client computes and uploads a weight update $\Delta w_{i,j}^{(t)}$ to anchor $j$.
9:     Anchor $j$ aggregates updates from all clients and updates its model:
10:         $W_j^{(t)} \leftarrow W_j^{(t-1)} + \eta_{\text{agg}} \sum_{i=1}^{N_{clients}} \frac{|C_i|}{|C|} \Delta w_{i,j}^{(t)}$
11: **end for**

12: **Stacking**
13: Generate meta-features from predictions of the final trained anchor models $\{W_j^{(T)}\}_{j=1}^{M_{anchors}}$.
14:     $P(x) \leftarrow [\text{Prob}(W_1^{(T)}, x), \ldots, \text{Prob}(W_{M_{anchors}}^{(T)}, x)]$
15: Train the stacking meta-model $S$ using the meta-features $P(x)$.
16: Perform final prediction using the trained meta-model $S$.
17:     Prediction $\leftarrow S(P(x))$.

---

### 3.3 TIME SERIES DATA DISTRIBUTION SHIFTS

The proposed Algorithm 2 detects changes in the data distribution of edge devices in a FL environment and maintains the frozen encoder in the latest state through cooperation between multiple anchors. When a learning round begins, each anchor server broadcasts the model weights for the current round to all connected edge devices. Each edge then estimates the current data distribution from the latest local dataset and compares it with the previous data distribution stored at the previous round. The difference between the two distributions is calculated using the Wasserstein distance, and if this value exceeds the predefined threshold of 0.1, a significant change is noted in the data environment. When a distribution change is detected, the edge notifies its anchor that migration to another anchor is necessary. Upon receiving this signal, the anchor communicates with other anchors within the network, comparing the representative data distributions with the latest distribution of the edge in question. Based on the comparison results, the anchor with the smallest distribution distance is selected as the target, and the SYN stage is initiated. The latest local model weights for the edge and, if necessary, some local data or statistics are transmitted to execute the actual migration. The SYN-ACK phase follows, where the target anchor combines the received model weights with its own model weights to create a new frozen encoder to return to the original anchor. Finally, in the ACK phase, the received encoder and the anchor encoder are updated to frozen encoders as a core

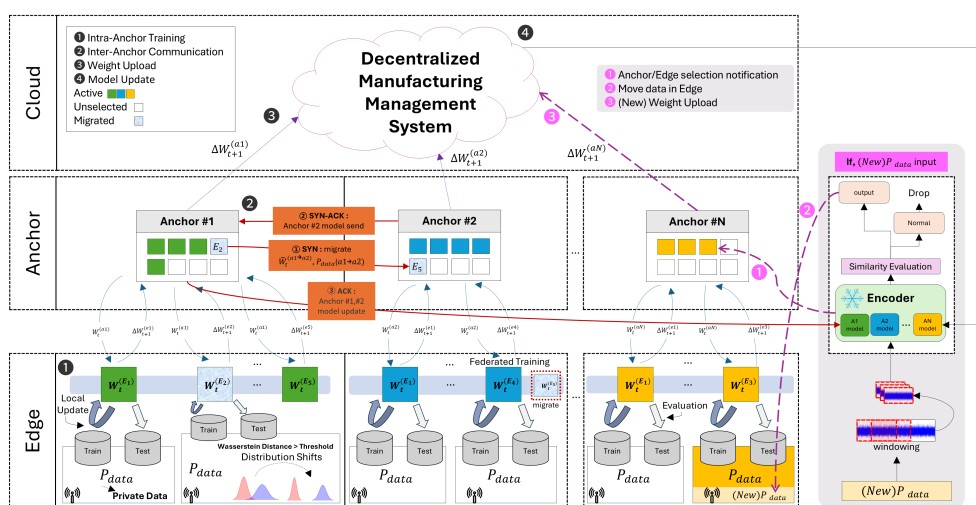

Figure 1: Proposed Decentralized Manufacturing Management System.

component for feature extraction of future edge data. These data are updated when the distribution changes or when the final round ends.

---

**Algorithm 2** Anchor-Based Frozen Encoder Update with Distribution Shift

---

1: **procedure** FEDERATEDLEARNINGROUND
   **1. Edge-level Training and Distribution Check**
2:     **for** each edge client $E_j$ **do**
3:         Train local model and compute a weight update.
4:         Compute current data distribution $P_{\text{current}}$ and load previous distribution $P_{\text{prev}}$.
5:         **if** Wasserstein distance $W_1(P_{\text{current}}, P_{\text{prev}}) > \tau$ **then**
6:             **SYN:** Send local model weights to a new target anchor.
7:         **end if**
8:     **end for**

   **2. Anchor-level Frozen Encoder Update**
9:     **for** each anchor $A_{target}$ that receives a model from an edge **do**
10:         **SYN-ACK:** Merge the received edge model with its own model.
11:         **ACK:** Update its frozen encoder with the merged model's knowledge.
12:         Propagate the updated frozen encoder to the edge's original anchor.
13:     **end for**
14: **end procedure**

---

### 3.4 NEW TIME SERIES DATA INPUT

This system utilizes a pre-trained frozen encoder with fixed weights to extract key features from input data, compares these features with predefined representative features of normal data, and evaluates relative similarity to determine anomalies. After determination, normal data are discarded or stored locally, while anomalous data are transmitted to the most suitable anchor for subsequent learning and analysis. When new data are received, windowing is performed to standardize the length and structure. In this process, continuously collected sensor signals are divided into fixed-length sequences and converted into a standardized form suitable for model input. The preprocessed data are passed to a frozen encoder consisting of a 1D convolutional layer, batch normalization, ReLU activation, global average pooling, and a fully connected layer. All parameters are fixed, enabling the encoder to reliably produce feature vectors for new data without further training. In the similarity evaluation stage, the extracted feature vectors are compared with the pre-calculated average feature

vectors for each anchor. The similarity between the input data and each anchor is quantified, and abnormality is determined based on relative similarity rather than an absolute distance threshold.

# 4 EXPERIMENTAL RESULTS

## 4.1 EXPERIMENTAL SETTING

In this study, we conduct two experiments to comprehensively verify the efficiency and performance of the proposed FL-based anomaly detection system using the stacking technique. The first experiment compares ensemble models. Here, various ensemble methods and regression-based models, such as XG-Boost, LightGBM, RandomForest, CatBoost, and Logistic Regression, are applied to evaluate prediction performance. The second experiment compares performance based on model structure in centralized learning and FL environments. The centralized learning environment includes traditional machine learning models such as Random Forest and XGBoost, while the FL environment independently applies various deep learning architectures such as CNN, GRU, LSTM, TCN, Transformer, and 1D-CNN. This allows analysis of inference time and classification accuracy. In comparison of FL algorithms, all use 100 learning rounds.

The dataset is partial discharge data for electrical fire accident prevention in industrial equipment, provided by AI-Hub. We used only csv files from multimodal data. The main purpose of the data was to predict and diagnose partial discharge, which is a major cause of electrical fires in industrial equipment. The data originate from nine power facilities (TFR-CV, CNCV-W, ACSR-OC, single-phase/power/instrumentation input transformers, 7.2kV/22.9kV switchgear, and 25.8kV GIS) corresponding to solid, liquid, and gas insulators and are categorized into normal, noise, surface discharge, corona discharge, and void discharge. The training data account for 80% of the set, the validation data for 10%, and the test data for 10%. The original data consist of 7,680 time series data points from 20 channels for a single partial discharge event. To improve the efficiency of model training, a method was applied to extract statistical characteristics of mean, standard deviation, maximum value, minimum value, skewness, and kurtosis from the time series data of each channel. That is, there are 5 anchors and 9 edges. Prior to the experiment, comparative experiments were conducted on various federated learning algorithms to select the optimal one. The algorithms compared were FedAdam, FedAvg, FedNova, FedProx, FedYogi, and Scaffold. FedProx($mu$=0.01) achieved a high macro F1 score of 0.74. As a consequence, the federated learning algorithm used FedProx($mu$=0.01) for the two experiments being performed. An ensemble technique was used with Xgboost.

## 4.2 COMPARISON OF ENSEMBLE MODELS

We compared and analyzed the performance of various ensemble models, which aim to overcome the limitations of a single model by combining the predictions of multiple models. TABLE 1 compares the overall F1 scores of the ensemble models. The LightGBM, random forest, and XGBoost models achieved the best performance, with a Macro F1 score of 0.75, which is higher than those of CatBoost and logistic regression. These results show that tree-based ensemble models learned more effectively than simple models such as logistic regression, considering the complex patterns.

Fig. 2 is a confusion matrix showing the classification results of each ensemble model. LightGBM, random forest, and XGBoost models clearly classify major classes of normal, noise, corona, surface, and void. The confusion matrices of these models have high values concentrated on the diagonal, suggesting a low misclassification rate.

Table 1: F1 Scores for Each Ensemble Model.

| Model | Accuracy | Macro F1 | Weighted F1 |
|---|---|---|---|
| CatBoost | 0.71 | 0.70 | 0.70 |
| LightGBM | 0.75 | 0.75 | 0.75 |
| Logistic Regression | 0.63 | 0.63 | 0.62 |
| Random Forest | 0.76 | 0.75 | 0.75 |
| XGBoost | 0.75 | 0.75 | 0.75 |

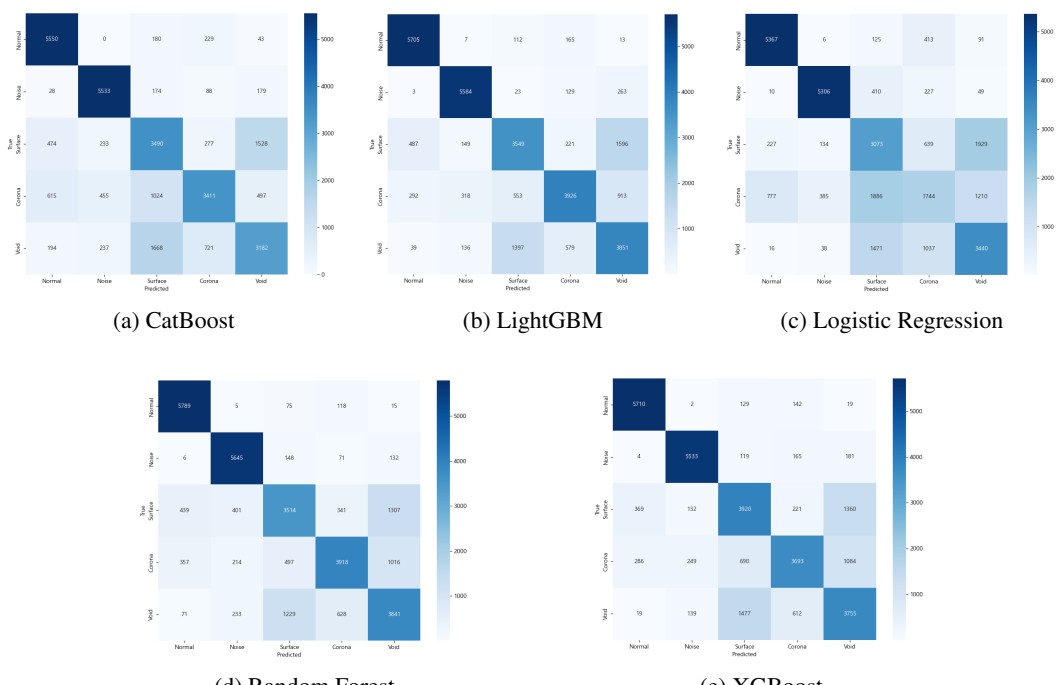

(a) CatBoost  (b) LightGBM  (c) Logistic Regression

(d) Random Forest  (e) XGBoost

Figure 2: Confusion Matrices for Each Ensemble Model.

### 4.3 COMPARISON OF CENTRALIZED TRAINING AND TYPICAL FEDERATED LEARNING

We compared and analyzed the performance of centralized learning, general FL, and FL models applying model stacking. TABLE 2 shows the comprehensive performance metrics of various models. Centralized learning demonstrated high performance because it utilizes all data. In particular, the LightGBM model performed excellently, recording an accuracy of 0.8618 and a Macro F1 of 0.8614. However, centralized learning has a clear limitation in that it requires access to all data, preventing the guarantee of data privacy. The FL+stacking model achieved an accuracy of 0.7438 and a Macro F1 score of 0.7424. This is significantly higher than the scores of individual models of standard FL and is close to the performance of centralized learning. The inference time was 2.3882 ms, which is longer than standard FL but significantly faster than centralized models. Standard FL recorded relatively lower performance compared to centralized models. Among these, the transformer model performed the best, with an accuracy of 0.6525 and a Macro F1 of 0.6516. This hinders FL, which learns locally across multiple edges, from achieving the same performance as centralized models. However, in terms of inference time, all models in standard FL were much more efficient than the centralized model.

TABLE 3 compares the F1 scores by anchor. The LightGBM model of centralized learning showed high performance in major anchors such as normal and noise but relatively low scores in the surface and corona anchors. The Transformer model in standard FL recorded high F1 scores in the normal and noise anchors. However, its performance was low in anchors of surface and void, suggesting that correction is needed for anchor imbalance. The FL+stacking model achieved high F1 scores in major anchors such as normal and noise. It also demonstrated much more stable and higher performance than the individual models of standard FL in the surface, corona, and void anchors.

TABLE 4 shows the performance by edge. The nine edge devices, from Edge 0 to 8, are comprised of the following equipment: TFR-CV, CNCV-W, ACSR-OC, single-phase/power/instrumentation input transformers, 7.2kV/22.9kV switchgear, and 25.8kV GIS. Centralized learning maintained high performance overall, but due to differences in data distribution by edge, there were cases of performance similar to or lower than other models in some edges. In the edge-specific performance, the transformer model of standard FL recorded a high F1 score of 0.6492 on Edge 3, but its performance was low on some edges such as Edge 1, indicating that it is affected by edge-specific data hetero-

geneity. The edge-specific performance also demonstrates the robustness of the FL+stacking model. Edge 7 recorded a very high F1 score of 0.9371, achieving results comparable to those of centralized learning. This demonstrates that stacking-based FL is effective in overcoming edge-specific data heterogeneity and stably improving overall performance.

Table 2: Accuracy and Inference Time Comparison: Centralized, Standard FL, and Stacking-based FL.

| Metrics | Centralized Learning | | | | Standard Federated Learning | | | | | | Federated Learning +Stacking |
|---|---|---|---|---|---|---|---|---|---|---|---|
| | Random Forest | XGBoost | LightGBM | CatBoost | CNN | GRU | LSTM | TCN | Transformer | 1D-CNN | 1D-CNN +Stacking |
| Accuracy | 0.8707 | 0.873 | 0.8618 | 0.8548 | 0.5236 | 0.5896 | 0.6819 | 0.5741 | 0.6525 | 0.5585 | 0.7438 |
| Macro F1 | 0.8702 | 0.8727 | 0.8614 | 0.8536 | 0.4645 | 0.5813 | 0.6764 | 0.5356 | 0.6516 | 0.5126 | 0.7424 |
| Weighted F1 | 0.8702 | 0.8727 | 0.8614 | 0.8536 | 0.4645 | 0.5813 | 0.6764 | 0.5356 | 0.6516 | 0.5126 | 0.7424 |
| Inference Time (ms) | 14.9524 | 0.266 | 0.913 | 0.464 | 0.231 | 0.1206 | 0.118 | 0.581 | 0.4192 | 0.242 | 2.3882 |

Table 3: Per-Anchor Performance Comparison: Centralized, Standard FL, and Stacking-based FL.

| Anchor | Centralized Learning | | | | Standard Federated Learning | | | | | | Federated Learning +Stacking |
|---|---|---|---|---|---|---|---|---|---|---|---|
| | Random Forest | XGBoost | LightGBM | CatBoost | CNN | GRU | LSTM | TCN | Transformer | 1D-CNN | 1D-CNN +Stacking |
| Normal | 0.9679 | 0.9649 | 0.966 | 0.951 | 0.6784 | 0.7459 | 0.8653 | 0.8557 | 0.8324 | 0.7945 | 0.9437 |
| Noise | 0.9434 | 0.9467 | 0.9454 | 0.9299 | 0.7901 | 0.8785 | 0.9138 | 0.7249 | 0.9459 | 0.8174 | 0.9216 |
| Surface | 0.8119 | 0.8088 | 0.7933 | 0.7786 | 0.529 | 0.3906 | 0.5048 | 0.5843 | 0.4959 | 0.5346 | 0.6188 |
| Corona | 0.813 | 0.8286 | 0.8104 | 0.8229 | 0.3142 | 0.3583 | 0.6426 | 0.2713 | 0.5206 | 0.3249 | 0.6885 |
| Void | 0.8151 | 0.8146 | 0.792 | 0.7854 | 0.0109 | 0.5333 | 0.4555 | 0.242 | 0.4633 | 0.0915 | 0.5395 |

Table 4: Per-Edge Performance Comparison: Centralized, Standard FL, and Stacking-based FL.

| Edge | Centralized Learning | | | | Standard Federated Learning | | | | | | Federated Learning +Stacking |
|---|---|---|---|---|---|---|---|---|---|---|---|
| | Random Forest | XGBoost | LightGBM | CatBoost | CNN | GRU | LSTM | TCN | Transformer | 1D-CNN | 1D-CNN +Stacking |
| Edge 0 | 0.7465 | 0.771 | 0.7678 | 0.6862 | 0.6261 | 0.757 | 0.7094 | 0.5891 | 0.7317 | 0.6073 | 0.5323 |
| Edge 1 | 0.8568 | 0.7975 | 0.7827 | 0.7792 | 0.2818 | 0.3752 | 0.3329 | 0.2311 | 0.6843 | 0.2235 | 0.2302 |
| Edge 2 | 0.7911 | 0.7629 | 0.7528 | 0.7677 | 0.4924 | 0.6565 | 0.6499 | 0.4282 | 0.7124 | 0.4474 | 0.44 |
| Edge 3 | 0.8291 | 0.8493 | 0.8443 | 0.8634 | 0.3559 | 0.6754 | 0.7822 | 0.6663 | 0.6492 | 0.6642 | 0.7705 |
| Edge 4 | 0.7499 | 0.7891 | 0.7358 | 0.8088 | 0.3109 | 0.2779 | 0.4536 | 0.5486 | 0.3288 | 0.4353 | 0.4701 |
| Edge 5 | 0.9736 | 0.9684 | 0.9526 | 0.9672 | 0.4985 | 0.5142 | 0.7301 | 0.6093 | 0.7076 | 0.4951 | 0.54 |
| Edge 6 | 0.99 | 0.9868 | 0.9776 | 0.8877 | 0.7269 | 0.7726 | 0.9203 | 0.7046 | 0.7487 | 0.7172 | 0.8369 |
| Edge 7 | 0.9552 | 0.9964 | 0.9968 | 0.9976 | 0.6994 | 0.7251 | 0.7852 | 0.5492 | 0.7833 | 0.7139 | 0.9371 |
| Edge 8 | 0.9212 | 0.9226 | 0.9245 | 0.8748 | 0.2221 | 0.3499 | 0.5683 | 0.4178 | 0.4224 | 0.3664 | 0.4568 |

## 5 CONCLUSION

This study proposes an DMMS based on federated learning using a three-layer structure of cloud–anchor–edge to overcome the limitations of data privacy, communication efficiency, and real-time analysis in manufacturing sites. In this structure, the edge protects raw data locally and performs learning, while the anchor aggregates models received from multiple edges and trains specialized models. Sensitive data are not shared externally, and each layer has a specified role, resulting in both scalability and security. To accurately identify complex anomalies that are difficult to capture with a single model, we applied a stacking ensemble technique that combines the prediction results of multiple anchor models trained through FL. Each anchor learns a model specialized for a specific type of anomaly, and the meta model re-learns their predictions to derive a final conclusion, enabling higher prediction accuracy and robustness.

Based on the results of this study, future research is needed in several areas. First, we plan to strengthen the generalization performance of the proposed system by applying and verifying the experiments using various datasets such as those of vibration, temperature, and pressure generated in actual manufacturing processes. Additionally, considering the limited computing resources of edge devices, we plan to maximize efficiency by applying additional lightweight models with the 1D-CNN model or introducing on-device learning optimization techniques such as quantization and pruning.

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

## A  DATASET AND PREPROCESSING

TABLE 5 is a summary of the partial discharge types, insulator types, and power equipment names.

Table 5: Classification of partial discharge types by power equipment insulator.

| Partial discharge type | Insulator type | Power equipment name |
|---|---|---|
| Normal | | TFR-CV |
| Noise | Solid | CNCV-W |
| Surface discharge | | ACSR-OC |
| Corona discharge | | Single-phase oil-filled transformer |
| Void discharge | Liquid | Power oil-filled transformer |
| | | Instrument transformer |
| | | 7.2kV switchgear |
| | Gas | 22.9kV switchgear |
| | | 25.8kVGIS |

The original data consist of 7,680 time series data points from 20 channels for a single partial discharge event. To improve the efficiency of model training, a method was applied to extract statistical characteristics of mean, standard deviation, maximum value, minimum value, skewness, and kurtosis from the time series data of each channel. These statistical features effectively summarize the core distribution and shape information of the data, enabling distinction of partial discharge types. To maximize the efficiency of processing numerous CSV files, we introduced a parallel processing technique based on multiprocessing. We divided the entire data file into multiple chunks and assigned them to multiple CPU cores for simultaneous processing, greatly reducing the data preprocessing time. The feature data and label information extracted from each chunk are stored in temporary files and then merged into a single integrated dataset. Through this process, the one-dimensional time series data are converted into a two-dimensional feature array in the form of *(20, 6)*, and this dataset is used as input for model training. After preprocessing, Edge 0 has 21,332 samples, Edge 1 has 21,373 samples, Edge 2 has 21,322 samples, Edge 3 has 21,332 samples, Edge 4 has 21,309 samples, Edge 5 has 21,341 samples, Edge 6 has 15,994 samples, Edge 7 has 16,063 samples, and Edge 8 has 31,918 samples.

## B  SUPPLEMENTARY EXPERIMENTS

The experimental environment used in this study is as follows. The main hardware and software specifications are summarized in TABLE 6. The CPU used is the Intel Core i7-13700F, a high-performance multi-core processor suitable for complex computations and data processing. The GPU

is an NVIDIA GeForce RTX 5060 Ti, which enables fast training and inference of deep learning models. The memory comprises 64GB of RAM, which supports largescale data processing and complex algorithm execution. The operating system is Windows 10 Pro, and the experiments were conducted in a Python 3.12.10 environment. Pytorch 2.7.1 was used for model development and learning optimization, and experiments were performed using CUDA version 12.8.

Table 6: Experimental environment hardware and software.

| Hardware Environment | Software Environment |
|---|---|
| CPU : Intel Core i7-13700F | OS : Windows 10 Pro |
| GPU : NVIDIA GeForce RTX 5060 Ti | Python : 3.12.10 |
| RAM : 64GB | PyTorch : 2.7.1 |
|  | CUDA : 12.8 |

In this study, we conduct four experiments to comprehensively verify the efficiency and performance of the proposed FL-based anomaly detection system using the stacking technique. The first experiment compares the anomaly detection performance of six representative FL algorithms—FedAvg, FedProx, Scaffold, FedNova, FedAdam, and FedYogi—under the same dataset and conditions. The convergence characteristics and accuracy changes of each algorithm are analyzed under data distribution imbalance and client communication delay. FedProx and Scaffold demonstrate imbalance mitigation effects, FedNova shows correction performance with partial client participation, and FedAdam and FedYogi compare the effects of adaptive optimizers. The second experiment compares performance based on model structure in centralized learning and FL environments. The centralized learning environment includes traditional machine learning models such as Random Forest and XG-Boost, while the FL environment independently applies various deep learning architectures such as CNN, GRU, LSTM, TCN, Transformer, and 1D-CNN. This allows analysis of inference time and classification accuracy. The third experiment compares ensemble models. Here, various ensemble methods and regression-based models, such as XG-Boost, LightGBM, RandomForest, CatBoost, and Logistic Regression, are applied to evaluate prediction performance. This allows analysis of model effectiveness for improving anomaly detection accuracy within the FL framework and determination of the most advantageous model depending on the data characteristics. The fourth experiment compares frozen encoder-based feature similarity metrics of Euclidean distance, cosine similarity, Pearson correlation coefficient, Minkowski distance, Chebyshev distance, Jaccard similarity, and Manhattan distance by calculating the distance between feature vectors extracted through the frozen encoder. We analyze the classification boundary characteristics and performance differences by metric in anomaly detection and identify the metric that has the highest correlation with the feature representation generated by the frozen encoder. These results serve as a basis for deciding the similarity metric to apply in future systems.

In this study's comparison of federated learning algorithms, all algorithms use 100 learning rounds, 1 local epoch, a batch size of 64, a learning rate of *1e-5*, and the Adam optimizer. To analyze the performance of the FedProx federated learning algorithm, three values of $\mu$ were used: *0.01*, *0.1*, and *1*. This $\mu$ value serves as a penalty coefficient to prevent the local model updates of each client from deviating significantly from the global model. The larger the value, the stronger the restriction on local updates, thereby enhancing model stability. The appropriate choice of $\mu$ depends on the characteristics of the data distribution and the system environment. In this study, we experimented with three values to evaluate their impact on algorithm performance. However, unlike other algorithms, the Scaffold algorithm sets the learning rate to 0.1 and uses SGD as the optimization method. The FedAdam and FedYogi algorithms set the server-side learning rate to 0.01 or 0.005, with $\beta_1$ set to 0.9, $\beta_2$ was set to *0.999*, and epsilon was set to *1e-8*. These settings allowed us to compare the learning efficiency and anomaly detection performance of each algorithm under the same data distribution and environment. In the ensemble model comparison, the following hyperparameters were set for each model. XGBoost was set to *use_label_encoder=False*, *eval_metric='mlogloss'*, *objective='multi:softprob'*, with the number of classes set to the same as the number of classes in the dataset, and *random_state=42* fixed. LightGBM was set to *objective='multiclass'*, *num_class* was set to the same value, the learning rate was set to *0.1*, *n_estimators=100*, and *random_state=42* was used. RandomForest was set to *random_state=42*, *n_estimators=100*, and the maximum depth (*max_depth*) was not limited. CatBoost was set with *loss_function='MultiClass'*, *iterations=100*, *learning_rate=0.1*, *depth=6*, *random_state=42*, and *verbose=False* to suppress logging during train-

ing. Logistic Regression was set with *max_iter=1000* and *random_state=42*. These fixed hyperparameters were used to control the training process of each ensemble model and enable performance comparison under the same conditions.

# C  TECHNICAL DETAILS

## C.1  FEDERATED LEARNING ALGORITHM

- FedAvg: FedAvg is the most fundamental algorithm in federated learning. Each client trains a model using local data, then transmits the model's weights to a central server. The server creates a new global model by taking the simple average of all weights sent by clients, and redistributes this to the clients (Reddi et al., 2021).

- FedAdam: FedAdam is an algorithm that applies the principles of the Adam optimizer when aggregating model updates on the server. Unlike FedAvg, which uses simple averaging, FedAdam applies momentum and adaptive learning rates to each client's update, improving the model's convergence speed and enhancing stability (McMahan et al., 2017).

- FedNova: FedNova is an algorithm designed to address client drift issues arising from data heterogeneity. Before aggregating each client's local updates, it normalizes the update norm to reduce bias caused by differences in learning rates or training iterations across clients, thereby promoting stable convergence (Wang et al., 2020).

- FedProx: FedProx is an algorithm for handling data heterogeneity. It adds a proximal regularization term to the loss function during local learning. This regularization term forces the client's local model to stay close to the global model, effectively mitigating client drift during training (Li et al., 2020).

- Fedyogi: Fedyogi is an algorithm that applies the Yogi optimizer, a variant of the Adam optimizer, to federated learning. Similar to FedAdam, it uses an adaptive optimization scheme when aggregating client model updates on the server. However, it induces more stable convergence by controlling the mean squared error of updates—a unique feature of Yogi—to prevent unstable updates. It particularly effectively prevents model divergence during training in heterogeneous data environments (Sattler et al., 2020).

- SCAFFOLD: SCAFFOLD is an algorithm that addresses client drift by introducing the concept of a control variable. When each client updates its local model, it uses a correction term to adjust for the difference between the local data and the global model. This guides each client's update direction to better align with the global optimization goal, significantly improving convergence speed and accuracy in heterogeneous data environments (Karimireddy et al., 2020).

  Translated with DeepL.com (free version)

## C.2  AI MODEL

- CatBoost: CatBoost is a gradient descent-based decision tree model developed by Russia's Yandex. Its most significant feature is its ability to efficiently handle categorical variables internally. Unlike existing models that required complex preprocessing to handle categorical variables, CatBoost automatically resolves this during training using a unique method called Ordered Boosting. This allows it to be applied directly to the data without separate transformations and also helps effectively prevent overfitting (Dorogush et al., 2018).

- LightGBM: LightGBM is a lightweight gradient descent boosting model developed by Microsoft. Instead of the level-based approach where trees grow horizontally, as in traditional boosting models, this model uses a leaf-based approach that prioritizes growing the leaf that maximizes loss reduction. This approach enables faster construction of more complex model structures. Additionally, it employs various optimization techniques, such as data parallelism and feature parallelism, resulting in a major advantage: extremely fast learning speeds even with large datasets (Ke et al., 2017).

- XGBoost: XGBoost is currently one of the most widely used gradient descent boosting models. It excels not only in performance but also in stability. It effectively controls model overfitting by inherently incorporating L1 and L2 regularization techniques. Additionally,

its built-in handling of missing values and support for various parallel processing techniques make it highly useful and scalable for managing complex real-world industrial data (Chen & Guestrin, 2016).

- LogisticRegression: Logistic Regression is the most fundamental statistical model used for classification problems. This model is based on a linear combination of the data, which is then passed through a sigmoid function to ultimately convert it into a probability value between 0 and 1. It predicts the likelihood of belonging to a specific category based on this probability value. Thanks to its simplicity and interpretability, it is frequently used as a baseline model to try first before applying more complex models (Stoltzfus, 2011).

- RandomForest: RandomForest is a representative ensemble model that combines multiple decision trees for prediction. This model uses a method where each tree is built by randomly sampling a portion of the data and randomly selecting features. It synthesizes the prediction results from these multiple trees to derive the final outcome. This approach effectively addresses the overfitting weakness inherent in a single decision tree, offering the advantage of more stable prediction performance (Breiman, 2001).

- CNN: CNN is a neural network optimized for processing two-dimensional or three-dimensional data such as images and videos. The core of this model is the Convolution layer, which extracts local features from the data. Filters scan the data to recognize features, generating feature maps based on them. Subsequently, the data size is reduced through pooling layers, and finally, a fully connected layer is used to perform the final classification or prediction (LeCun et al., 1998).

- GRU: GRU is a type of recurrent neural network used for processing time-series data. It was developed to overcome the limitation of traditional RNNs forgetting past information due to the vanishing gradient problem in long sequence data. GRU uses two gates—an update gate and a reset gate—to autonomously decide which information to retain and which to discard. It features a simpler structure than LSTM while delivering comparable performance, making it efficient for use in environments with relatively limited data or computational resources (Cho et al., 2014).

- LSTM: Alongside GRU, LSTM is the most representative model for solving the long-term dependency problem in recurrent neural networks. It manipulates data through three gates—a more complex input gate, a forget gate, and an output gate—along with an internal cell state. This complex structure enables it to effectively remember and utilize important past information, even in very long sequence data, demonstrating outstanding performance (Hochreiter & Schmidhuber, 1997).

- TCN: TCN is a model that processes sequence data using only convolution operations, abandoning the recursive structure of recurrent neural networks. This model employs causal convolution and dilated convolution to secure a wide receptive field, enabling it to view a broad range of input data at once while referencing only past information. Unlike RNNs, this allows for parallel processing, resulting in extremely fast learning speeds (Bai et al., 2018).

- Transformer: The Transformer is a model that processes sequence data using only the Attention mechanism, without the recursive/convolution structures of RNNs or CNNs. Its core component, Self-Attention, calculates the relationships between all elements within the input sequence, allowing each element to determine its own importance relative to others. This capability allows it to fully understand context, driving innovation in the NLP field and making it one of the most widely used models today (Vaswani et al., 2017).

- 1D-CNN: 1D-CNN is a convolutional neural network specialized for one-dimensional data like time series or text. Similar to how 2D CNNs move across images, the filters in a 1D-CNN move only in one direction, extracting local patterns from the data. For example, in time-series data, it effectively recognizes patterns of signal sequences of a specific length, while in text, it identifies patterns of word groups within sentences (Kiranyaz et al., 2016).

## C.3  SIMILARITY EVALUATION

- Manhattan Distance: Also known as the L1 norm, the Manhattan distance is a metric that measures the distance between two points when traveling only along grid paths. It is named

as such because it resembles moving only east, west, south, and north along city blocks. This distance is calculated by summing the absolute differences in each dimension. Unlike Euclidean distance, it possesses useful characteristics for creating models that are less sensitive to environments with many obstacles or outliers.

- Minkowski Distance: Minkowski distance is an Lp norm, a generalized distance measurement method that encompasses both Manhattan and Euclidean distances. The calculation method varies depending on the value of the parameter p. When p = 1, it is equivalent to the Manhattan distance; when p = 2, it is equivalent to the Euclidean distance. Setting p to infinity yields the Chebyshev distance. Because it encompasses various distance metrics with a single formula, it is used when one wishes to flexibly define distance according to the characteristics of the data.

- Euclidean Distance: Also known as the L2 norm, Euclidean distance is the most common method for measuring the shortest straight-line distance between two points. It is a multidimensional extension of the Pythagorean theorem in a two-dimensional plane, calculated by squaring the difference in each dimension, summing them, and taking the square root. This metric is widely used in many machine learning algorithms, such as K-Means Clustering and K-NN, to measure similarity between data points.

- Chebyshev Distance: Chebyshev distance defines the distance as the maximum absolute difference among multiple differences between two points. It is equivalent to the maximum distance a king can move in a single turn in chess, whether horizontally, vertically, or diagonally. In parallel computing, when waiting for all tasks of multiple processes to complete, the time taken by the slowest task equals the total time, making this metric useful for modeling such scenarios.

- Jaccard Similarity: Jaccard similarity is a metric expressing the similarity between two sets as a percentage. It is calculated as the size of the intersection divided by the size of the union. For example, it is used to calculate the percentage of items purchased in common by two users or to measure the percentage of words shared between two documents to determine similarity. This metric is unaffected by the size of the two sets, making it particularly useful when the presence or absence of data is critical.

- Cosine Similarity: Cosine similarity measures how closely two vectors point in the same direction. It ignores the magnitude of the vectors and considers only the angle between them. This makes it highly effective for high-dimensional data, such as documents where word frequencies are represented as vectors. Even if document lengths differ, similar relative word importance or usage patterns result in high similarity.

- Pearson Correlation: The Pearson correlation coefficient is a measure of the strength and direction of a linear relationship between two variables. This value always ranges between -1 and 1, where 1 indicates a perfect positive linear relationship, -1 indicates a perfect negative linear relationship, and 0 indicates no linear relationship. Since it is unaffected by changes in the data's scale or mean, it is very useful for determining how closely the trends of two variables align.

## D    DETAILED EXPERIMENTAL RESULTS

### D.1    COMPARISON OF FEDERATED LEARNING ALGORITHMS

In this study, we compared the performance of various FL aggregation algorithms using data. TABLE 7 shows the F1 scores for each aggregation algorithm. Based on the F1 score, the algorithm with the best performance was FedYogi, which recorded a high F1 score of 0.78 in both cases. Next, FedAdam (learning rate 0.01) showed decent performance with a score of 0.77. On the other hand, FedNova and Scaffold showed relatively low F1 scores, which may be due to the non-homogeneity (non-IID) of the data and instability during the model convergence process. Despite these results, this study selected FedProx ($\mu$=0.01) as the final model, with an F1 score of 0.75, equivalent to FedAvg, showing a respectable result that was not significantly different from that of the best-performing model. The FedProx algorithm focuses on mitigating data heterogeneity issues by adding a regularization term to local updates. Fig. 3 is a graph comparing the accuracy, loss, and F1-score of each aggregation algorithm over 100 rounds. FedProx ($\mu$=0.01) showed a very stable convergence curve

during the learning process. The legend shows the various FL algorithms compared in this study and the main hyperparameter settings for each model. FedProx_1, FedProx_0.1, FedProx_0.01 refer to cases where the proximal term coefficient $\mu$ in the FedProx algorithm is set to 1, 0.1, and 0.01, respectively; FedAdam_0.005 and FedAdam_0.01 represent cases with a server learning rate in the FedAdam algorithm set to 0.005 and 0.01, respectively. FedYogi_0.01 and FedYogi_0.005 refer to cases where the server learning rate of the FedYogi algorithm is 0.01 or 0.005, respectively. FedAvg, FedNova, and Scaffold are models that apply the standard settings for each algorithm.

The F1 score curve shows slightly lower final performance compared to FedYogi or FedAdam, but it did not exhibit the unstable variability seen in FedNova or Scaffold. TABLE 8 shows the performance of each algorithm by edge node. Each facility was sequentially assigned edge numbers from 0 to 8. Edge 0 represents TFR-CV; Edge 1 represents CNCV-W; Edge 2 represents ACSR-OC; and Edges 3, 4, and 5 represent single-phase incoming transformers, power-use incoming transformers, and meter-use incoming transformers, respectively. Edges 6 and 7 are 7.2kV and 22.9kV distribution panels, respectively; and Edge 8 is a 25.8 kV GIS facility. FedYogi and FedAdam show relatively small performance variations between edge nodes, indicating that all clients participated stably in learning. On the other hand, FedProx showed relatively large performance variations, with high performance on some edge nodes and low performance on others.

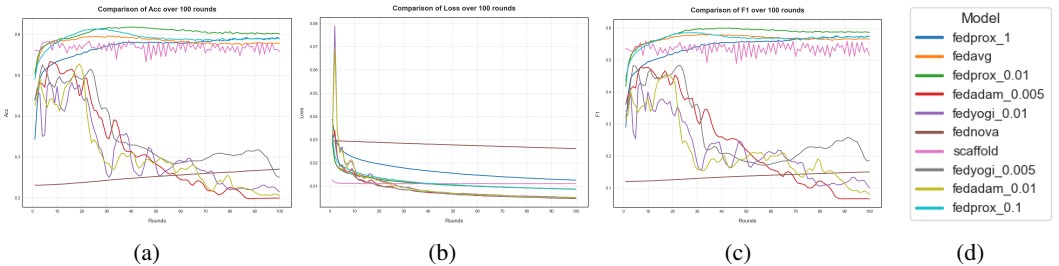

|     (a)     |     (b)     |     (c)     |     (d)     |

Figure 3: Comparison of Accuracy, Loss, and F1 Score by the Aggregation Algorithm over 100 Rounds: (a) accuracy, (b) loss, (c) F1-score, and (d) legend.

Table 7: F1 Scores for Each Aggregation Algorithm.

| Algorithm | F1 | Macro F1 | Weighted F1 |
|---|---|---|---|
| Fedadam_0.01 | 0.77 | 0.76 | 0.76 |
| Fedadam_0.005 | 0.76 | 0.76 | 0.76 |
| Fedavg | 0.75 | 0.75 | 0.75 |
| Fednova | 0.63 | 0.63 | 0.63 |
| Fedprox_0.01 | 0.75 | 0.75 | 0.75 |
| Fedprox_0.1 | 0.74 | 0.74 | 0.74 |
| Fedprox_1 | 0.74 | 0.74 | 0.74 |
| Fedyogi_0.01 | 0.78 | 0.78 | 0.78 |
| Fedyogi_0.005 | 0.78 | 0.78 | 0.78 |
| Scaffold | 0.73 | 0.73 | 0.72 |

Table 8: Performance Comparison of Federated Learning Algorithms by Edge Node.

| Algorithm | Edge 0 | Edge 1 | Edge 2 | Edge 3 | Edge 4 | Edge 5 | Edge 6 | Edge 7 | Edge 8 |
|---|---|---|---|---|---|---|---|---|---|
| Fedadam_0.01 | 0.1073 | 0.0667 | 0.0667 | 0.0669 | 0.0672 | 0.0667 | 0.1166 | 0.0855 | 0.1393 |
| Fedadam_0.005 | 0.0667 | 0.0667 | 0.0667 | 0.0667 | 0.0667 | 0.0667 | 0.0667 | 0.0667 | 0.0667 |
| Fedavg | 0.524 | 0.1841 | 0.4401 | 0.8279 | 0.5136 | 0.5628 | 0.9361 | 0.957 | 0.3111 |
| Fednova | 0.1886 | 0.1077 | 0.0989 | 0.2066 | 0.2445 | 0.1895 | 0.1249 | 0.1622 | 0.0749 |
| Fedprox_0.01 | 0.5486 | 0.2617 | 0.4755 | 0.7644 | 0.5201 | 0.5666 | 0.8241 | 0.9856 | 0.3528 |
| Fedprox_0.1 | 0.5087 | 0.2272 | 0.4269 | 0.7066 | 0.4949 | 0.555 | 0.9013 | 0.9792 | 0.3593 |
| Fedprox_1 | 0.5302 | 0.1824 | 0.5066 | 0.8129 | 0.5924 | 0.5291 | 0.9342 | 0.8656 | 0.3247 |
| Fedyogi_0.01 | 0.1444 | 0.068 | 0.1396 | 0.0739 | 0.0696 | 0.0664 | 0.1005 | 0.0684 | 0.0667 |
| Fedyogi_0.005 | 0.0697 | 0.0667 | 0.2678 | 0.1791 | 0.0786 | 0.0 | 0.5409 | 0.433 | 0.0667 |
| Scaffold | 0.4292 | 0.1884 | 0.3423 | 0.7235 | 0.494 | 0.7102 | 0.7807 | 0.676 | 0.4277 |

## D.2 ENSEMBLE MODEL PERFORMANCE

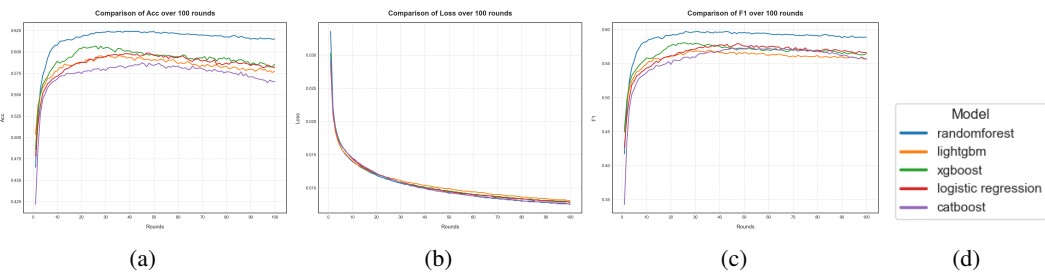

(a)        (b)        (c)        (d)

Figure 4: Comparison of Accuracy, Loss, and F1 Score by Ensemble Model over 100 Rounds: (a) accuracy, (b) loss, (c) F1-Score, and (d) legend.

Table 9: Macro F1 Score by Edge for Each Ensemble Model.

| Model | Edge 0 | Edge 1 | Edge 2 | Edge 3 | Edge 4 | Edge 5 | Edge 6 | Edge 7 | Edge 8 |
|---|---|---|---|---|---|---|---|---|---|
| CatBoost | 0.5765 | 0.2013 | 0.4890 | 0.7454 | 0.5254 | 0.4764 | 0.8709 | 0.9884 | 0.2503 |
| LightGBM | 0.5143 | 0.2438 | 0.4154 | 0.7824 | 0.4963 | 0.5089 | 0.8496 | 0.9558 | 0.3359 |
| Logistic Regression | 0.5896 | 0.2320 | 0.4745 | 0.8065 | 0.5122 | 0.4998 | 0.8456 | 0.9511 | 0.3644 |
| Random Forest | 0.5630 | 0.1993 | 0.4654 | 0.7927 | 0.5538 | 0.5630 | 0.9113 | 0.9972 | 0.3456 |
| XGBoost | 0.6239 | 0.2652 | 0.4277 | 0.8278 | 0.5275 | 0.6181 | 0.7958 | 0.9647 | 0.3220 |

The models included in the comparison were CatBoost, LightGBM, logistic regression, random forest, and XGBoost. Fig. 4 shows the changes in accuracy, loss, and F1 score for each ensemble model at each learning round. TABLE 9 is the Macro F1 scores recorded by ensemble model for each edge. Fig. 4 shows the learning stability and convergence trends of the ensemble models. LightGBM, random forest, and XGBoost showed rapid performance improvement in the early stages of learning and then converged stably while maintaining the highest accuracy and F1 scores for 100 rounds. In contrast, logistic regression and CatBoost showed relatively lower performance curves compared to these models. CatBoost showed a trend of steady performance improvement as learning progressed, despite its low initial performance. TABLE 9 is an important indicator showing the results of ensemble models with data heterogeneity for each edge. Performance varied by edge, but the patterns differed by model.

## D.3 SIMILARITY EVALUATION RESULTS

We compared various similarity evaluation metrics and analyzed the characteristics and limitations of each metric. Predictions were based on the first sample of five randomly selected test samples, and the results are shown in TABLE 10. Fig. 5 shows the t-SNE results, where the data points for each class do not form distinct clusters but are mixed together. This suggests ambiguous boundaries between classes and a complex data distribution, complicating accurate determination of similarity using a single metric. TABLE 10 shows the similarity evaluation results. The cosine similarity

metric evaluated the similarity to the predicted class as high (0.9391), while the similarity to the correct class was evaluated as low (0.5093). This indicates that cosine similarity, which measures similarity of vector direction, cannot clearly distinguish prediction errors in this dataset. On the other hand, Euclidean distance measured the difference from the predicted class as 3.2969 and the difference from the correct class as 7.1019, reflecting the prediction error. The Pearson correlation coefficient showed a positive correlation with the predicted class and a negative correlation with the correct class, effectively identifying the prediction error. This result suggests that Euclidean distance or Pearson correlation coefficient may provide more meaningful prediction error information than cosine similarity in cases where the boundaries between classes are ambiguous.

Table 10: Similarity Evaluation Metrics.

| Metric | True Label | Predicted Label | Normal | Noise | Surface | Corona | Void |
|---|---|---|---|---|---|---|---|
| Manhattan Distance | Corona | Normal | 24.3757 | 66.4178 | 61.2881 | 63.0805 | 63.3553 |
| Minkowski Distance | Corona | Normal | 1.8861 | 3.7431 | 3.7764 | 3.9643 | 3.6184 |
| Euclidean Distance | Corona | Normal | 3.2969 | 7.3216 | 7.1019 | 7.3983 | 6.9915 |
| Jaccard Similarity | Corona | Normal | 0.5859 | 0.5859 | 0.5859 | 0.5859 | 0.5859 |
| Cosine Similarity | Corona | Normal | 0.9391 | 0.5271 | 0.5772 | 0.5093 | 0.5847 |
| Chebyshev Distance | Corona | Normal | 1.1359 | 1.6274 | 1.7437 | 1.8426 | 1.4422 |
| Pearson Correlation | Corona | Normal | 0.8946 | 0.0264 | 0.0514 | -0.0181 | 0.0713 |

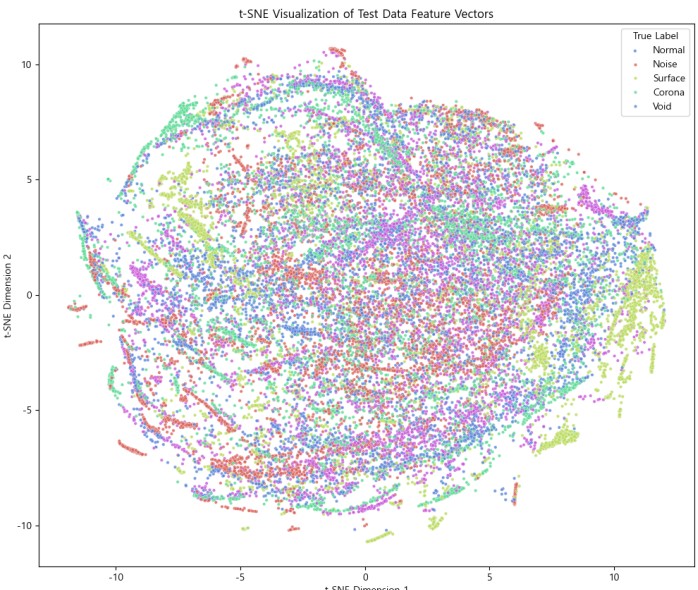

Figure 5: t-SNE Visualization of the AI-Hub Dataset.

