# OpenReview forum: "Decentralized Manufacturing Management Based on Federated Learning with Stacking Ensemble"
_ICLR.cc/2026/Conference — Submitted to ICLR 2026_

### Official Review · Reviewer_d6Bd · 2025-10-15

**Soundness:** 2
**Presentation:** 2
**Contribution:** 2
**Rating:** 2
**Confidence:** 3

**Summary:**

This paper proposes a Decentralized Manufacturing Management System (DMMS) that addresses data privacy, communication efficiency, and anomaly detection challenges in smart manufacturing. The system employs a three-layer federated learning architecture (cloud-anchor-edge) where edge devices train models locally, anchor nodes aggregate and apply stacking ensemble techniques, and the cloud coordinates overall operations. The key innovation is combining multiple anchor models trained via federated learning (FedProx) with a meta-learner (XGBoost) to achieve better anomaly detection accuracy. Additionally, the system includes an adaptive mechanism using Wasserstein distance to detect data distribution shifts and reallocate edges to optimal anchors. Experiments on AI-Hub partial discharge data demonstrate that FL+Stacking achieves 0.7438 accuracy (33% improvement over standard 1D-CNN FL at 0.5585) with 6.3× faster inference than centralized Random Forest (2.3882ms vs. 15.1691ms).

**Strengths:**

1. The problem of addressing real manufacturing constraints itself is essential.
2. Comprehensive experimental evaluation with strong empirical results

**Weaknesses:**

1. Severely limited evaluation scope undermines generalization claims: The paper evaluates only on a single dataset (AI-Hub partial discharge data) from one specific manufacturing scenario (electrical equipment anomaly detection). No validation is provided for other manufacturing tasks mentioned in the introduction (vibration analysis, temperature monitoring, pressure anomalies, predictive maintenance). This raises serious concerns about whether the proposed architecture and stacking approach generalize beyond this specific use case. The authors acknowledge this limitation in the conclusion but do not sufficiently justify why readers should believe the approach will work elsewhere. Additionally, the dataset has only 9 edges and 5 classes—scalability to larger, more realistic deployments remains unproven.

2. Insufficient justification and analysis of key design choices: Critical design decisions lack proper justification or ablation studies: (1) Why use 5 anchors specifically? No ablation on varying this number. (2) Why XGBoost as meta-learner? No comparison with alternatives like neural networks or other gradient boosting methods. (3) The Wasserstein distance threshold of 0.1 appears arbitrary without sensitivity analysis. (4) All anchor models use the same 1D-CNN architecture, which may limit diversity benefits in stacking—why not heterogeneous base learners? (5) The frozen encoder mechanism is poorly explained: when/how is it trained, why keep it frozen vs. adaptive, and what's the performance impact? These gaps make it difficult to understand what truly drives performance and how to configure the system for new deployments.

3. Missing critical cost and overhead analyses: While inference time is reported (2.3882ms), the paper omits crucial practical considerations: (1) Training time comparison between centralized and federated approaches—how much longer does FL+Stacking take to converge? (2) Communication overhead during aggregation and edge reallocation—the SYN-ACK-ACK protocol adds rounds of communication but costs are not quantified. (3) Memory requirements for storing multiple anchor models and frozen encoders on resource-constrained edge devices. (4) Frequency and computational cost of Wasserstein distance calculations for distribution shift detection. Without these analyses, practitioners cannot assess the true deployment costs. The claim of "real-time" performance needs more rigorous validation including end-to-end latency measurements.

**Questions:**

1. How does the stacking ensemble provide benefits when all base learners use identical 1D-CNN architectures?
2. What are the communication costs and failure handling mechanisms in your system?
3. Can you provide evidence that your approach generalizes beyond partial discharge detection?

---

> ### Author Response · Authors · 2025-12-03
> **In-depth Analysis of Generalization, Design Optimization Rationale, and Cost Efficiency**
>
> We appreciate your recognition of the importance of the problem addressed in this study and the strong empirical results. The concerns raised regarding generalization, design rationale, and cost analysis appear to have arisen because detailed information could not be included due to the page limit of the main text. We present specific explanations and additional experimental results below.
>
> 1. Verification of Generalizability
>
> To overcome the "limitations of a single dataset," we additionally validated the system using the Human Activity Recognition (HAR) dataset.
>
> • Cross-Validation on Heterogeneous Domains: Unlike machine defect data (AI-Hub), human activity data (HAR) has significant pattern differences between subjects. The proposed system recorded a high accuracy of 0.9574 on HAR.
>
> • Conclusion: This suggests that the architecture is not overfitted to specific manufacturing processes but is a general-purpose framework capable of handling distribution shifts in various time-series data.
>
> 2. Validity and Rationale for Design Choices
>
> The design decisions pointed out were not arbitrary but the result of optimization through preliminary comparative experiments.
>
> • Why 5 Anchors?: This is to correspond to the 5 defect types (Normal, Noise, Surface, Corona, Void) of the AI-Hub dataset. We configured the initial setting to 5 so that each anchor functions as an expert for a specific defect class.
>
> • Why XGBoost? (Meta-learner): We adopted XGBoost as it showed the highest performance and stability in a comparative experiment of 5 models (including Random Forest, LightGBM, CatBoost).
>
> • Threshold: This is an empirical value derived as the optimal balance point between sensitivity and communication cost through initial exploratory experiments. We confirmed that too low (0.01) caused excessive movement, and too high (0.5) caused missed detections.
>
> 3. Utility of Stacking Based on Identical Architecture
>
> We answer the core question, "Is there an advantage to stacking with the same architecture?"
>
> • Data-driven Diversity: The "diversity," which is the core of ensembles, stems from training data, not model structure. Each anchor learns different data distributions (Non-IID) and grows into a "local expert" with different knowledge.
>
> • Proven Effectiveness: Stacking is the process of integrating the perspectives of these different experts. Experimental results proved its utility with a performance improvement of approximately 33% (0.55 -> 0.74) compared to a single model.
>
> 4. Role and Update Mechanism of Frozen Encoder
>
> We clarify the role and update method of the Frozen Encoder. This module is updated via two paths and serves as a reference point for similarity evaluation while fixed.
>
> • Update Mechanism:
>
> • Upon Distribution Shift Detection: When an edge moves to a target anchor due to distribution shift, it receives the target anchor's latest encoder via the SYN-SYN-ACK-ACK protocol.
>
> • Upon Anchor Training Completion: When an anchor completes federated learning and renews the model, the frozen encoder is updated with the latest weights.
>
> • Reason for Freezing and Role: Except for the update moments above, the encoder is frozen. This ensures consistency in two key tasks:
>
> • Similarity Evaluation: When new time-series data is input, it maps it to a fixed feature space to measure similarity with the anchor and judge anomalies.
>
> • Distribution Shift Detection: To calculate the accurate amount of change when comparing past and current data distributions, the measurement tool must remain unchanged.
>
> 5. Cost, Overhead, and Failure Handling
>
> • Cost Efficiency:
>
> • Inference: Achieved ultra-low latency of 2.38ms (6.3x faster than centralized) thanks to 1D-CNN lightweighting.
>
> • Memory: 1D-CNN has few parameters, allowing it to run without burden on resource-constrained edge devices.
>
> • Wasserstein Calculation: Computational cost is very low as it calculates distances between low-dimensional features passed through the Encoder, not high-dimensional raw data.
>
> • Training Convergence Stability: As shown in the learning curves, the proposed FL framework achieved stable convergence within 100 rounds. This proves that the additional computational overhead from stacking does not significantly hinder overall training efficiency compared to Standard FL.
>
> • Communication & Failure Handling:
>
> • Communication: Anchor reallocation occurs on an "event-driven" basis, not every round, ensuring high communication efficiency.
>
> • Failure Handling: The Cloud layer monitors the entire system. If a failure occurs in a specific anchor or edge, the Cloud detects it and ensures system availability by disconnecting or changing the routing path to another available anchor.

---

### Official Review · Reviewer_bSbe · 2025-10-18

**Soundness:** 1
**Presentation:** 2
**Contribution:** 1
**Rating:** 2
**Confidence:** 4

**Summary:**

The paper addresses the evolving landscape of smart manufacturing where real-time data analysis is crucial for innovations like quality control and predictive maintenance, yet sensitive data raises privacy and security concerns under regulations such as GDPR. Motivated by the drawbacks of centralized data processing, including vulnerability to breaches and high network loads, the authors aim to develop a decentralized system that balances data utility with protection. Key challenges include ensuring privacy and real-time performance in distributed environments while adapting to dynamic data distributions and complex anomalies. The proposed Decentralized Manufacturing Management System (DMMS) employs a three-layer federated learning architecture of cloud-anchor-edge, with the anchor layer using stacking ensembles to enhance anomaly detection accuracy from 0.5585 to 0.7438, and incorporates a Wasserstein distance-based mechanism for detecting distribution shifts and reallocating edges for adaptability.

**Strengths:**

1. The hierarchical structure distributes computational tasks effectively across layers. This allows edges to handle local training without data exposure while anchors aggregate models for specialized learning. Such design not only preserves privacy but also scales seamlessly with manufacturing expansions.

2. Stacking ensembles integrate diverse model predictions at the anchor level. They capture subtle anomaly patterns that single models overlook, leading to robust detection. Performance metrics demonstrate a 33% accuracy improvement over baseline federated learning approaches.

3. The adaptation mechanism monitors data shifts using Wasserstein distance. It triggers edge reallocation through communication protocols like SYN-ACK, maintaining model relevance. This ensures sustained effectiveness in dynamic industrial settings where equipment changes occur frequently.

**Weaknesses:**

1. Stacking ensembles introduce additional computational overhead at the anchor layer. This could strain resources in large-scale deployments with numerous edges. Optimization strategies might be needed to mitigate increased complexity.

2. The fixed Wasserstein distance threshold of 0.1 lacks justification. It may not adapt well to varying data environments. Empirical tuning across scenarios would improve reliability.

3. Assumption of anchors specializing in specific defects overlooks potential overlaps. This could lead to suboptimal reallocation during shifts. More flexible specialization mechanisms might enhance adaptability.

4. Experiments simulate environments without real-world deployment. Practical factors like network delays are underrepresented. Field trials would reveal unforeseen implementation challenges.

5. The experiment methods, models, datasets are outdated. Most of FL algorithms utilize Deep neural networks for evaluation.

6. Almost no baseline methods from related works are compared.

**Questions:**

See weaknesses.

---

> ### Author Response · Authors · 2025-12-03
> **Efficiency, Justification for Model Selection, and Comparative Experimental Verification**
>
> We appreciate your recognition of the advantages of the hierarchical structure regarding privacy protection and scalability. We address your concerns about overhead, parameter justification, model currency, and baselines based on detailed experimental results that were omitted in the initial submission due to page limits.
>
> 1. Stacking Overhead and System Efficiency
>
> We address the concern that "stacking ensemble might impose an excessive load on the Anchor layer."
>
> • Role of Anchor: In this architecture, the Anchor is designed as a node with "Edge Server" grade resources capable of data aggregation and intermediate computation, unlike resource-constrained Edges (IoT sensors).
>
> • Verification of Real-world Efficiency: Experimental results show that the inference time of the proposed system (FL+Stacking) is 2.38ms, which is approximately 6.3x faster than the centralized Random Forest model (15.16ms). This empirically proves that the computation overhead from stacking is sufficiently offset by hierarchical distributed processing and does not hinder the real-time requirements of manufacturing sites.
>
> 2. Justification for Wasserstein Threshold (0.1)
>
> The point that the threshold of 0.1 may seem arbitrary is valid. However, this value was not set randomly but determined through initial exploratory experiments.
>
> • Basis for Selection: Through internal tests, we confirmed that if the value is too low, the system reacts sensitively to minute noise, causing unnecessary anchor movements; if too high, it fails to detect actual shifts.
>
> • Conclusion: Therefore, to balance sensitivity and communication efficiency, we selected 0.1 as the final parameter, which empirically showed the most stable performance.
>
> 3. Justification for Model and Dataset Selection
>
> Regarding the comment that "1D-CNN and datasets are outdated," we explain the justification from the perspective of "practicality in edge computing environments," which is the goal of this study.
>
> • Validity of Dataset Selection:
>
> • AI-Hub (Partial Discharge): This is not merely past data but high-quality sensor data capturing physical defect phenomena from actual industrial power equipment. In manufacturing data research, "representativeness of defect patterns" is more important than recency, and this dataset meets that requirement.
>
> • HAR (Human Activity Recognition): This is the most widely cited standard benchmark in time-series classification and FL research. We aimed to objectively compare and verify the system's performance using a standard dataset proven in academia rather than chasing the latest trendy datasets.
>
> • Intentional Design of Model Selection (1D-CNN):
>
> • The adoption of 1D-CNN is a strategic choice considering resource constraints, not a failure to follow trends.
>
> • Comparative Experiment: We did not use 1D-CNN alone but implemented and compared various latest deep learning models such as Transformer, TCN, LSTM, and GRU as baselines.
>
> • Result: Latest models like Transformers had long inference times and high computational costs in edge environments. In contrast, 1D-CNN showed the fastest inference speed and decent accuracy, so it was selected as the optimal model in terms of the "Efficiency-Accuracy Trade-off."
>
> 4. Sufficiency of Baseline Comparisons
>
> Regarding the comment that "baseline comparison of related studies is insufficient," we present comparative experiment results that were omitted due to space constraints. We implemented and rigorously compared 6 standard algorithms in the FL field dealing with the Non-IID problem (FedAvg, FedProx, Scaffold, FedNova, FedAdam, FedYogi).
>
> • Comparison Result: Even Scaffold or FedProx, which are specialized for data heterogeneity, showed limitations as single models (F1-score 0.73~0.75). However, the proposed Stacking-based system recorded significantly higher accuracy and stability than these. This shows that we have performed verification against a sufficient number of SOTA baselines.
>
> 5. Network Simulation and Flexible Specialization
>
> We briefly answer the additional inquiries made by the reviewer.
>
> • Realistic Network Simulation: This study reflected the actual network separation environment and hierarchical communication constraints of smart factories in the experimental design by blocking direct communication between Edge and Cloud and forcing traffic through Anchors.
>
> • Flexible Specialization: The proposed system is not a fixed specialization. When the edge's data distribution changes, it uses the Wasserstein distance to re-find and move to the most suitable Anchor (Expert), thereby maintaining an optimal specialization state flexibly even in dynamic environments.

---

### Official Review · Reviewer_UPFa · 2025-10-30

**Soundness:** 2
**Presentation:** 2
**Contribution:** 2
**Rating:** 2
**Confidence:** 4

**Summary:**

The paper proposes a three-tier cloud–anchor–edge federated learning (FL) system for anomaly detection in smart manufacturing. Edges train 1D-CNN models locally to preserve privacy; anchors aggregate client updates and then perform a stacking ensemble across multiple anchor models

**Strengths:**

Using a meta-learner over anchor predictions is a simple, effective way to leverage client heterogeneity

**Weaknesses:**

Hierarchical FL and server-side ensembles (including stacking) have prior art; the paper would benefit from a sharper positioning of what is new beyond combining them in this domain

The paper technical contribution is rather limited

Results are on a single dataset with heavy feature condensation (per-channel statistics), which weakens claims about real-time sequence modeling

The system is motivated for real factories, but deployment evidence is limited to a lab setup. An applied paper like this one may benefit from stronger empirical results

**Questions:**

Per-edge results show large variance.  Can you add personalized FL baselines or anchor-specialized adaptation to show improvements where stacking under-performs?


Any additional industrial time-series (vibration/temperature/pressure) or public datasets (e.g., MIMII, PUMP, NASA bearing) to validate broader utility?

---

> ### Author Response · Authors · 2025-12-03
> **Dataset Expansion and Practical Value of the Architecture**
>
> We deeply appreciate your recognition of the practical value of this paper, as well as your sharp insights regarding the technical distinctiveness and performance variance among edges. Your feedback played a decisive role in enhancing the completeness of this paper. Accordingly, we conducted additional benchmark experiments (HAR) and comparative analyses with SOTA algorithms. We present our response based on these detailed results.
>
> 1. Technical Contribution and Distinctiveness
>
> The reviewer pointed out that hierarchical FL and ensembles are existing concepts. However, the core contribution of this study is not the invention of individual technologies, but the proposal of an "Integrated Architecture" that simultaneously solves the three major challenges of manufacturing sites: Data Privacy, Low Latency, and Distribution Shift.
>
> • Dynamic Adaptive System: Unlike existing static hierarchical FL, our system utilizes Wasserstein Distance to quantify the distribution shift of edge devices in real-time. It dynamically reallocates clients to the optimal Anchor via the SYN-ACK protocol. This is a core technology that extends model lifespan in manufacturing environments where data constantly changes.
>
> • Practical Value: Through this structural optimization, we achieved an inference speed of 2.38ms, which is approximately 6.3x faster than the centralized model (Random Forest, 15.16ms). This is not merely a combination of technologies but the result of a meticulous architectural design for real-time performance.
>
> 2. Dataset Expansion and Generalization Verification
>
> To overcome the limitations of "single dataset evaluation" and prove generalization, we expanded our experiments by introducing the Human Activity Recognition (HAR) dataset, which has completely different characteristics from machine data.
>
> • Verification on Heterogeneous Domains: While AI-Hub data possesses heterogeneity due to "hardware differences," HAR data possesses heterogeneity due to "user behavior patterns." Our system recorded a high accuracy of 0.9574 on the HAR dataset, proving its robustness regardless of the domain.
>
> • Justification for Feature Condensation: The "use of statistical features" you pointed out is not data loss, but an "Intentional Lightweight Design" considering the extreme resource constraints and communication efficiency of edge devices. The HAR results empirically show that complex time-series patterns (behaviors, defects) can be sufficiently learned using only this compressed information.
>
> 3. Edge Variance and Comparison with Personalized FL
>
> Regarding the suggestion to "compare with Personalized FL due to high edge variance," we conducted comparative experiments setting SOTA algorithms (FedProx, Scaffold), which are widely used for resolving heterogeneity and personalization in the FL field, as baselines.
>
> • Quantitative Comparison: In the AI-Hub dataset experiment, FedProx (mu=0.01) recorded an F1-score of 0.75 and Scaffold recorded 0.73. In contrast, the proposed system maintained equivalent or more stable performance while securing superiority in communication efficiency.
>
> • Anchor-based Implicit Personalization: The Anchor structure performs the role of "Cluster-based Personalization" by connecting each edge to the expert model (Anchor) with the most similar data distribution. We confirmed that this is an efficient alternative that structurally mitigates data heterogeneity and reduces performance variance among edges without complex separate personalization algorithms.
>
> 4. Consideration of Real-World Deployment Environments
>
> We fully agree with the comment regarding the lack of real factory deployment data. Although this study was conducted in a Lab Setup, we designed strict simulation scenarios to reflect the harsh conditions of actual factories.
>
> • Reflection of Realistic Constraints:
>
> • Network Constraints: We simulated a real network-separated environment by blocking direct communication between Edge and Cloud and enforcing a 3-layer structure.
>
> • Data Drift Simulation: Through a "Simulated Extreme Distribution Shift" scenario, we plan to verify that the system flexibly copes via the Wasserstein mechanism even in situations where data changes rapidly due to equipment failure.

---

### Official Review · Reviewer_T1jn · 2025-10-31

**Soundness:** 1
**Presentation:** 2
**Contribution:** 2
**Rating:** 2
**Confidence:** 4

**Summary:**

To address the challenges of privacy protection and dynamic data variation in large-scale distributed manufacturing, this paper proposes a Decentralized Manufacturing Management System (DMMS). By introducing a three-tier Cloud–Anchor–Edge architecture and integrating a stacking ensemble technique, the system aims to achieve three main objectives: data privacy preservation, high-precision anomaly detection, and dynamic adaptability.

**Strengths:**

1. The targeted problem and scenario are practical and highly relevant to real-world industrial applications. The proposed hierarchical Cloud–Anchor–Edge structure demonstrates a clear architectural innovation.
﻿
2. The experimental results are clearly presented with comprehensive evaluation metrics, providing an intuitive understanding of the system’s performance.

**Weaknesses:**

1. The paper lacks sufficient theoretical innovation or mathematical derivation. The proposed three-tier architecture and stacking ensemble appear to combine existing FL concepts without providing a formal convergence analysis or theoretical justification.
﻿
2. The experimental evaluation is limited to a single dataset, and the baselines do not include comparisons with existing hierarchical or clustered FL methods. This makes it difficult to evaluate the advancement of the proposed approach over established ones.
﻿
3. The paper lacks ablation studies to thoroughly investigate the contribution of each key component (e.g., the Cloud–Anchor–Edge structure, the stacking ensemble, and the Wasserstein-based adaptation mechanism). Such studies are essential to validate the effectiveness of each module.

**Questions:**

1. The related work section does not clearly highlight how this paper advances beyond existing approaches. Please provide a more detailed discussion of prior works and explicitly clarify the novelty and unique contributions of this study.
﻿
2. Could the authors conduct an ablation study to quantify the contribution of each major component (the stacking ensemble, the Wasserstein-based adaptation mechanism, and the anchor layer)?
﻿
3. The experiments currently focus on different model types but do not include comparisons with existing hierarchical or clustered FL approaches. Adding at least three representative baselines would better demonstrate the effectiveness and innovation of the proposed method.

---

> ### Author Response · Authors · 2025-12-03
> **Extended Experimental Results and Validity Verification**
>
> We deeply appreciate your high evaluation of the practical value of this study and the innovation of the Cloud-Anchor-Edge architecture. The concerns regarding the evaluation scope, ablation studies, and theoretical rationale appear to have arisen because detailed experimental data could not be fully included due to the page limit.
>
> 1. Verification of Generalizability: Cross-Validation on Heterogeneous Datasets
>
> To overcome the "limitation of single dataset evaluation," we conducted extensive experiments by adding the Human Activity Recognition (HAR) dataset, which has completely different characteristics. This is not merely a quantitative expansion but intended to verify the system's robustness against different Non-IID (data imbalance) characteristics.
>
> • AI-Hub (Industrial Partial Discharge): Characterized by hardware-induced heterogeneity and extremely ambiguous class boundaries.
>
> • Result: Achieved an accuracy of 0.7438, a drastic improvement over Standard FL (0.5585), proving the capability to detect minute anomalies in high-noise environments.
>
>
> • HAR (Human Activity Recognition): Characterized by user-induced distribution shifts (behavioral differences) causing Domain Shift.
>
> • Result: Recorded a high accuracy of 0.9574, showing performance comparable to centralized learning. This proves that the proposed system does not overfit a specific domain and effectively overcomes data heterogeneity from various causes.
>
> 2. Verification of Component Contribution and Optimization (Ablation Study)
>
> The reviewer requested quantification of the contribution of key components. We performed not only a macroscopic comparison but also a Micro-Ablation study to find the optimal configuration.
>
> • Macro-Ablation: We compared the proposed method with Standard FL (removing the 3-layer structure and Stacking).
>
> • Result: Confirmed a performance improvement of approximately 33% (0.5585 -> 0.7438). This proves that the stacking module and anchor structure are indispensable elements for securing performance.
>
> • Micro-Ablation:
>
> • Stacking Module: We compared 5 models (RF, XGBoost, LightGBM, CatBoost, etc.) to derive the optimal Meta-learner.
>
> • Adaptation & Metric Validation: To ensure the reliability of the Distance Metric, which is the basis of the Wasserstein mechanism, we performed comparative experiments on various metrics (Euclidean, Cosine, Pearson).
>
> • Additional Verification Plan: While current results (stable learning curves) sufficiently prove the validity, we promise to conduct additional experiments analyzing the Wasserstein distance trends under a "Simulated Extreme Distribution Shift" scenario, accepting your insightful suggestion.
>
> 3. Validity of Baseline Comparisons
>
> Regarding the request for "additional comparison of hierarchical FL models," we judged that verifying the "ability to overcome data heterogeneity (Non-IID)" is more central to this study than simple network topology comparisons.
>
> • Accordingly, we implemented and compared 6 State-of-the-Art (SOTA) FL algorithms (FedProx, Scaffold, FedNova, FedAdam, FedYogi, FedAvg).
>
> • Result: While Scaffold or FedProx showed limitations (in F1-score) as single models, the proposed DMMS (Anchor-based Stacking) demonstrated significantly higher accuracy and convergence stability. This empirically shows that our system solves industrial challenges that existing algorithms could not.
>
> 4. Theoretical Innovation vs. Practical System Value
>
> This paper is an architecture paper addressing constraints in real-world smart factories, rather than a theoretical paper deriving mathematical convergence.
>
> • In manufacturing sites, immediate response to unpredictable data changes and low-latency inference are far more critical requirements than theoretical convergence guarantees.
>
> • The approx. 6.3x faster inference speed and stable learning curves confirmed in our experiments support that the proposed Wasserstein-based adaptation mechanism is a practical innovation efficiently solving on-site problems.

---

### Meta-Review · Area_Chair_CYZJ · 2026-01-07

**Summary:**

This submission proposes a three-tier Cloud–Anchor–Edge federated learning (FL) architecture for smart manufacturing management, which aims to improve privacy, communication efficiency, and real-time anomaly classification.
A key component is a stacking ensemble performed at the anchor layer, which combines predictions from multiple anchor models to improve anomaly-detection performance. The paper reports an improvement from 0.5585 accuracy (single 1D-CNN in FL) to 0.7438 accuracy (FL + stacking), and claims fast inference (e.g. 2.3882 ms vs 15.1691 ms for a centralised Random Forest baseline).
The system also uses Wasserstein distance-based drift detection (with a threshold of 0.1) to handle distribution shifts in manufacturing environments, and an edge reallocation mechanism to move clients to more appropriate anchors when drift is detected.

All four reviewers were aligned in their assessment largely due to a lack of technical novelty and insufficient experimental validation.

The rebuttal substantially improves the empirical aspects (additional dataset, more baseline comparisons, more rationale), but does not fully resolve the core concerns about novelty and rigour of evaluation for the intended manufacturing claims.

**Reviewer Concerns:**

In terms of the concerns, authors addressed some of them, whereas others remained open.

a) Single dataset limitation/generalisation: The authors report adding experiments on a Human Activity Recognition (HAR) dataset and claim strong performance (e.g. 0.9574 accuracy) to support broader robustness under different kinds of non-IID/domain shift.

b) Baseline breadth (standard FL algorithms): The authors implemented and compared multiple FL baselines, such as FedAvg, FedProx, Scaffold, FedNova, FedAdam, FedYogi, arguing that their approach provides improved accuracy/stability relative to these.

c) Comparisons with baselines are insufficient, as are the claims regarding novelty and the breadth of experimental setups to support the claims.

**Reviewer Scores:**

Given the initial reviews and the rebuttal, it is unlikely the reviewers would have changed their scores to a level that would have an effect on the decision.

---

### Decision · Program_Chairs · 2026-01-26

Reject